



# The sensitivity of the colour of dust in MSG-SEVIRI Desert Dust infrared composite imagery to surface and atmospheric conditions

Jamie R. Banks[1], Anja Hünerbein[1], Bernd Heinold[1], Helen E. Brindley[2], Hartwig Deneke[1], and Kerstin Schepanski[1]

[1]Leibniz Institute for Tropospheric Research (TROPOS), Leipzig, Germany.
[2]Space and Atmospheric Physics Group, and NERC National Centre for Earth Observation, Imperial College London, London, UK.

*Correspondence to:* Jamie R. Banks (banks@tropos.de)

**Abstract.**

Infrared "Desert Dust" composite imagery taken by the Spinning Enhanced Visible and InfraRed Imager (SEVIRI), onboard the Meteosat Second Generation (MSG) series of satellites above the equatorial East Atlantic, has been widely used for more than a decade to identify and track the presence of dust storms from and over the Sahara Desert, the Middle East, and southern

Africa. Dust is characterised by distinctive pink colours in the Desert Dust false-colour imagery, however the precise colour is influenced by numerous environmental properties, such as the surface thermal emissivity and skin temperature, the atmospheric water vapour content, and the quantity and height of dust in the atmosphere. This paper is the follow-up to Banks et al. (2018), which analysed the sensitivity of the colour of the dust in the imagery to its infrared optical properties. The previous paper introduced a modelling system combining dust concentrations simulated by the aerosol transport model COSMO-MUSCAT

(COSMO: COnsortium for Small-scale MOdelling; MUSCAT: MUltiScale Chemistry Aerosol Transport Model) with radiative transfer simulations from the RTTOV (Radiative Transfer for TOVS) model, in order to simulate the SEVIRI infrared measurements and imagery. Investigating the sensitivity of the synthetic infrared imagery to the environmental properties over a six month summertime period from 2011 to 2013, it is confirmed that water vapour is a major control on the apparent colour of dust, obscuring its presence when the moisture content is high. Of the three SEVIRI channels used in the imagery (8.7,

10.8, and 12.0 $\mu$m), the channel at 10.8 $\mu$m has the highest atmospheric transmittance and is therefore the most sensitive to the surface skin temperature. A direct consequence of this sensitivity is that the background desert surface exhibits a strong diurnal cycle in colour, with light blue colours possible during the day and purple hues prevalent at night. In dusty scenes, the clearest pink colours arise from high-altitude dust in dry atmospheres. Elevated dust influences the dust colour primarily by reducing the contrast in atmospheric transmittance above the dust layer between the SEVIRI channels at 10.8 and 12.0 $\mu$m,

thereby boosting red and pink colours in the imagery. Hence the higher the dust altitude, the higher the threshold column moisture needed for dust to be obscured in the imagery: for a sample of dust simulated to have an AOD at 550 nm of 2-3 at an altitude of 3-4 km, the characteristic colour of the dust may only be impaired when the total column water vapour is particularly moist ($\gtrsim$ 39 mm). Meanwhile dust close to the surface (altitude < 1 km) is only likely to be apparent when the atmosphere is particularly dry and when the surface is particularly hot, requiring column moistures $\lesssim$ 13 mm and skin temperatures $\gtrsim$ 314 K,

and is highly unlikely to be apparent when the skin temperature is $\lesssim$ 300 K. Such low altitude dust will regularly be almost



invisible within the imagery, since it will usually be beneath much of the atmospheric water vapour column. It is clear that the interpretation of satellite-derived dust imagery is greatly aided by knowledge of the background environment.

# 1 Introduction

Desert dust storms originating from the Sahara are known to have significant effects on climate, for instance via their radiative effects (e.g. Haywood et al., 2005; Allan et al., 2011; Ansell et al., 2014; Banks et al., 2014; Alamirew et al., 2018), but it is also clear that the climatic environment of the desert has significant effects on the generation mechanisms and the transport of atmospheric dust (e.g. Schepanski et al., 2009b; Wagner et al., 2016; Gasch et al., 2017; Schepanski et al., 2017; Schepanski, 2018). Dust activity is coupled with its environment, the one feedbacks on the other.

The fact that the environment influences dust storm activity may be intuitively obvious, but what may be less obvious is that the background environment also affects the capability of measuring and imaging dust activity from satellite observational data (e.g. Wald et al., 1998). Satellite instruments such as SEVIRI (Spinning Enhanced Visible and InfraRed Imager (Schmetz et al., 2002)) have been widely used over the past decade and a half as a powerful tool to both visualise and quantify the presence of atmospheric dust over desert regions (e.g. Schepanski et al., 2007; Ashpole and Washington, 2012), however the interpretation of these measurements does require some knowledge of the surface and atmospheric properties contributing to the signal detected by the satellite instrument (e.g. Ackerman, 1997; Legrand et al., 2001). For example, it is known that atmospheric moisture can hide the presence of dust especially in infrared (IR) measurements and imagery (Brindley et al., 2012), while other contributing factors to the measured IR signal include the surface thermal emissivity and the surface (skin) temperature, as well as various properties of the dust itself, for example its vertical distribution. Disentangling the signals of the atmospheric variability in the measurements from the variability of the dust signature is vital in order to assess the quantity, nature, and effects of the dust itself. In order to do this, it is necessary to understand the background signals in the satellite measurements. A more precise knowledge of the atmospheric and surface environment would greatly contribute to the interpretation of the satellite measurements and imagery.

This paper is the companion paper to a previous study (Banks et al., 2018), henceforth denoted B2018, which explored the influence of dust optical properties on the apparent colour of dust in summertime 'Desert Dust' infrared RGB (red-green-blue) composite imagery produced from SEVIRI measurements (Lensky and Rosenfeld, 2008). Dust in the 'Desert Dust' imagery displays characteristic pink colours, hence it is often also referred to as the 'Pink Dust' imagery. An example of a Saharan dust storm is presented in Figure 1, imaged using this scheme during the day and at night, depicting a large swathe of dust over the Algerian/Malian border. B2018 introduced a tool to simulate the SEVIRI Desert Dust imagery using the dust transport model COSMO-MUSCAT (COSMO: COnsortium for Small-scale MOdelling; MUSCAT: MUltiScale Chemistry Aerosol Transport Model) in concert with the radiative transfer model RTTOV (Radiative Transfer for TOVS). COSMO-MUSCAT is able to simulate the meteorological situation over North Africa as well as dust emissions, transport and deposition, while RTTOV uses this information to simulate the resultant radiances and brightness temperatures that would be measured at the top-of-the-atmosphere (TOA) by SEVIRI. The benefit of this technique is that the sensitivity of the SEVIRI measurements and imagery



to the spatial distributions of dust, to the IR optical properties of the dust, as well as to the surface and atmospheric conditions, may be tested. Comparing with daytime SEVIRI measurements, B2018 found that the synthetic Desert Dust imagery appears to be most sensitive to dust in a size bin with an effective radius of 1.5 $\mu$m: despite being more numerous, particles in smaller size ranges have rather weaker extinction properties, while particles in larger size ranges have much smaller concentrations. Various

IR optical property databases were tested, with that created by Volz (1973) appearing to be most consistent with the SEVIRI measurements. The sensitivity to particle shape was also explored, indicating that elongated spheroidal particles increase the interaction of the particles with IR radiation, but only if they display a tendency to be aligned horizontally, which may not be an appropriate assumption.

The previous study (B2018) explored the influences of the IR optical properties of the dust on the composite imagery, but left

open the questions as to the influences of the surface and atmospheric properties on the ability to perceive dust in the imagery. Following up on these open questions, this paper will therefore seek to explore the surface and atmospheric influences in greater detail, with reference to the surface thermal emissivity, the skin temperature, the atmospheric column moisture, and the dust altitude. Hence this paper will be structured as follows: Section 2 will provide background information as to the satellite observations, the COSMO-MUSCAT-RTTOV modelling system, and the assumed dust IR optical properties; Section 3 will

explain the meteorological situation over North Africa during summer, and explore the frequency of occurrence and the co-location of the variables; and finally Section 4 will provide a synthesis of the various factors, describing how the interlocking surface and atmospheric features give rise to characteristic colours in the imagery, with reference to the night-time situation and to the dust height, the latter being a vital control on the brightness temperature differences which govern the measurements underpinning the imagery.

## 20   2   Satellite observations and modelling strategy

This section provides a brief recap of the methods and modelling strategy presented by the companion paper, Banks et al. (2018): for more details and for a more substantial discussion on these, see Sections 2 and 3 therein.

### 2.1   Satellite imagery from MSG-SEVIRI

In geostationary orbit above the Gulf of Guinea since 2002 (located at 0°N, currently at 0°E), the Meteosat Second Generation

(MSG) series of satellites is operated by EUMETSAT (the European Organisation for the Exploitation of Meteorological Satellites) to maintain high-temporal resolution, real-time measurements of weather systems over Europe and Africa. Onboard the MSG satellites is the SEVIRI instrument (Schmetz et al., 2002), an imaging radiometer which measures the Earth-atmosphere system in 11 narrow-band visible and IR channels. A twelfth channel provides high spatial resolution broadband visible measurements. The temporal resolution is 15 minutes and the spatial sampling rate at nadir of the 11 narrow-band channels is 3 km.

Given its location, SEVIRI is ideally situated for taking continuous measurements of African weather, of which atmospheric desert dust is an important component due to the significant impact of dust storms on human health (e.g. Griffin, 2007) and human activities.





The three IR atmospheric window channels at 8.7, 10.8, and 12.0 $\mu$m are particularly useful for measurements and imaging of dust aerosol over desert regions, since clear-sky atmospheric transmission is high at these wavelengths: measurements at these wavelengths are therefore most representative of the apparent surface, whether that is land, ocean, cloud, or aerosol (e.g. dust). Meanwhile visible channels have greater difficulty in resolving dust aerosols over deserts due to the weak contrast in

5 reflectance between the desert and dust aerosols. SEVIRI IR measurements are often interpreted in the form of brightness temperatures $T_B$, considering the IR radiance measurements to be representative of blackbody radiation; see Schmetz et al. (2002) and Lensky and Rosenfeld (2008) for more details.

'Desert Dust' RGB composite imagery from SEVIRI is defined in the following configuration, as specified by Lensky and Rosenfeld (2008):

$$10 \quad R = \frac{(T_{B120} - T_{B108}) - R_{min}}{R_{max} - R_{min}} \qquad\qquad (R_{min} = -4\,K, R_{max} = +2\,K) \qquad\qquad (1)$$

$$G = \left(\frac{(T_{B108} - T_{B087}) - G_{min}}{G_{max} - G_{min}}\right)^{1/2.5} \qquad (G_{min} = 0\,K, G_{max} = 15\,K) \qquad\qquad (2)$$

$$B = \frac{T_{B108} - B_{min}}{B_{max} - B_{min}} \qquad\qquad (B_{min} = 261\,K, B_{max} = 289\,K). \qquad\qquad (3)$$

This formulation has been designed specifically to highlight the presence of dust aerosol in the imagery, making use of brightness temperatures at 10.8 $\mu$m in the blue beam and brightness temperature differences (BTDs) in the red and green beams.

RGB values have a range of 0 to 1, so for example if $T_{B108}$ has a value of 300 K, which is greater than the maximum value of 289 K, the blue value will saturate at 1. More details of the characteristic features present in this imagery will be described in Section 3, along with the conditions which give rise to them (see also B2018, Section 2.1).

## 2.2 COSMO-MUSCAT simulations of dust emissions and concentrations, and RTTOV simulations of SEVIRI composite imagery

The atmosphere-dust modelling system COSMO-MUSCAT is used to simulate 3-D atmospheric properties and dust aerosol concentrations over North Africa, as described by, e.g. Heinold et al. (2011) and Schepanski et al. (2017). It is composed of the coupling between version 5.0 of the atmospheric model COSMO (Schättler et al., 2014) and the chemistry tracer transport model MUSCAT (Wolke et al., 2012). Dust is considered as a passive tracer, and is described by 5 size bins with particle radii between 0.1 and 24 $\mu$m. COSMO-MUSCAT also includes online radiative feedback between the dust fields and solar and

thermal radiation (Helmert et al., 2007).

Considered here are COSMO-MUSCAT simulations from the Junes and Julys of 2011, 2012, and 2013. These months cover the periods of the Fennec campaign in 2011 and 2012 (Washington et al., 2012; Ryder et al., 2015), an aircraft and ground-based measurement campaign in western areas of the central Sahara, the SALTRACE (Saharan Aerosol Long-range TRansport and Aerosol-Cloud-interaction Experiment) campaign over the northern tropical Atlantic in 2013 (Weinzierl et al., 2017), and

30 the ChArMEx (Chemistry-Aerosol Mediterranean Experiment) campaign over the western Mediterranean (Mallet et al., 2016), also in 2013. MSG-2 was the operational MSG satellite during the summers of 2011 and 2012, while MSG-3 was operational in summer 2013. To cover both the North African desert regions and the possible Atlantic and European dust outflow regions,





the model domain extends from 0 to 60°N, and 30°W to 35°E. The horizontal grid spacing is 28 km, while vertically there are 40 sigma-$p$ levels starting with a bottom layer 20 m thick. Simulation output is produced at a 3-hourly resolution.

The radiative transfer model RTTOV has been designed for fast radiative transfer simulations of satellite measurements (Saunders et al. (1999), Matricardi (2005), publicly available at https://www.nwpsaf.eu/site/software/rttov/, last accessed 15th

November 2018). Taking surface, atmospheric, and aerosol properties as input, RTTOV calculates the radiances and brightness temperatures that would be measured at TOA, for a defined satellite instrument and channel, for example the SEVIRI channels at 8.7, 10.8, and 12.0 μm. Associated with these radiances, RTTOV also provides the atmospheric transmittance as output, defined as the fraction of the radiance emitted by the surface which passes out to space. Surface properties include thermal emissivity and skin temperature, while atmospheric properties include pressure, temperature, and specific humidity. Solar and

viewing zenith angles are also accounted for. Aerosol properties are defined by profiles of the absorption and scattering coefficients (km$^{-1}$) and of the back-scatter parameter, which is the integrated fraction of back-scattered energy (Matricardi, 2005). COSMO-MUSCAT-RTTOV is the modelling system which combines COSMO-MUSCAT simulations with the capabilities of RTTOV in simulating the SEVIRI brightness temperatures (Banks et al., 2018), so as to produce synthetic Desert Dust imagery. In order to simulate only dusty or clear scenes, RTTOV simulations are not performed for COSMO-MUSCAT grid cells with

cloud fractions greater than 1%. Meanwhile in order to focus on desert regions and the desert margins, RTTOV simulations are only performed in the latitude band between 12 and 36°N, over land.

## 2.3  Optical properties of dust

Knowledge of the optical properties of dust is required both to calculate the aerosol optical depth (AOD), and to calculate the IR effects of dust on the SEVIRI TOA brightness temperatures as simulated by RTTOV. Optical properties are derived from

the wavelength-dependent refractive indices of the dust type and the size and shape of the dust particles. Dust particles are assumed to be spherical, see B2018 for more analysis of the influences of particle shape on the imagery.

Optical properties are calculated using Mie theory (e.g. Mishchenko et al., 2002), which requires as input the complex refractive index $m(\lambda)$ and the scattering parameter $2\pi r_{\mathrm{eff}}$ / $\lambda$, dependent both on the wavelength $\lambda$ and the particle effective radius $r_{\mathrm{eff}}$. These inputs are used to calculate the extinction, absorption, and scattering efficiencies, which are convolved over

the SEVIRI channel filter response functions to calculate the SEVIRI channel efficiencies $Q(r_{\mathrm{eff}}, ch)$. Extinction coefficients, $\beta(r_{\mathrm{eff}}, ch, t, x)$, for each channel, location $x$, and time $t$, are calculated with reference to the particle size and density $\rho_{\mathrm{p}}$ (specified as that of quartz, 2.65 g cm$^{-3}$), and to the mass concentration $M$ in the grid box (e.g. Tegen et al., 2010).

The AOD is defined by the integration of the extinction coefficient over the entire atmospheric column. For consistency with satellite AOD retrieval products (e.g. Hsu et al., 2013), which commonly quote their AOD retrievals using visible wavelengths,

COSMO-MUSCAT dust AODs (Helmert et al., 2007) are calculated at 550 nm using dust visible refractive indices presented by Sinyuk et al. (2003). Validation of the North African COSMO-MUSCAT simulated AODs with respect to AOD retrievals from AERONET (Holben et al., 1998) has been performed and is described by B2018, Section 4.

In the IR, calculations of the SEVIRI channel extinction, absorption and scattering coefficients are carried out using refractive indices developed by Volz (1973), henceforth denoted VO73. Further optical properties (Sokolik and Toon, 1999; Hess et al.,



1998; Di Biagio et al., 2017) databases were also tested by B2018, however the VO73 refractive indices appeared to give the closest match to the observed AODs and the composite image colour, and so in this paper only COSMO-MUSCAT-RTTOV simulations with spherical VO73 dust will be considered. Summarising the optical properties relevant to this paper, Table 1 specifies the size-resolved extinction efficiencies for the five COSMO-MUSCAT size bins, at 550 nm for the Sinyuk et al.
(2003) dust, assumed to be spherical, and for the three SEVIRI IR channels of interest using VO73 dust. Specified also in Table 1 are the bin size ranges and effective radii.

## 3   The atmospheric and surface environment of North Africa

In Section 3.1 the meteorological background to the summertime North African environment is introduced, while Sections 3.2 to 3.5 consider the influences of the North African environment on the satellite IR measurements and imagery.

### 3.1   North African atmospheric dynamics

The summertime (June, July, August) Saharan climate is of particular research interest due to the complexity of the meteorological situation, giving rise to a number of different patterns of atmospheric dust generation and transport. During the summer, the atmospheric circulation over North Africa is predominantly characterised by the strength of the Harmattan north-easterly trade winds, the pulsating nature of the Saharan Heat Low (SHL, e.g. Engelstaedter et al. (2015)), and northward migration of
monsoonal air (Schepanski et al., 2017). The southerly monsoon winds, which bring humid and cooler air masses from the Gulf of Guinea into the continent, and the northerly Harmattan winds, which push dry and hot air masses southward, meet at the Inter-Tropical Discontinuity (ITD), a zone marked by a strong gradient in humidity and a jump in wind direction. Generally, dry and desert conditions are present over the North African continent north of the ITD, but inflows of cooler airmasses from the adjacent seas result in intermittent increases in humidity and reduced temperatures. Particular examples include Mediterranean
cold air surges (Vizy and Cook, 2009) and the inflow of marine air masses over Western Sahara (Grams et al., 2010).

Dust entrained into the planetary boundary layer (PBL) is vertically mixed and eventually homogeneously distributed over the depth of the PBL (e.g. Schepanski et al., 2009b). Most dust source activations occur during the first half of the day (Schepanski et al., 2017), in particular after sunrise when the PBL is deepening and the turbulent mixing increases. However over some parts of the Sahara and the Sahel in summer more than half of the dust emission can be related to moist convection
(haboobs) from late afternoon to early evening (Allen et al., 2013; Heinold et al., 2013). Whereas larger dust particles fall out quickly within a couple of hours, smaller particles remain aloft for longer (days to weeks). These particles will be transported away from the source region eventually leaving the continent. At sunset, the turbulent eddies determining the depth and the mixing within the daytime PBL decay and a calm and shallower nocturnal boundary layer (NBL) grows from the surface into the former daytime PBL, referred to as the residual layer. Whereas dust suspended in the NBL settles down due to the lack of
turbulent buoyancy, dust suspended in the residual layer may be transported efficiently across longer distances during the night (e.g. Kalu, 1979; Schepanski et al., 2009a).





## 3.2  Surface thermal emissivity

The surface thermal emissivity is one of the controlling variables on the pristine-sky IR composite imagery. In this context, 'pristine-sky' refers to cloud-free and aerosol-free scenes. The emissivity is a wavelength-dependent property, by definition bounded within a range between 0 and 1, but in practice for the three SEVIRI channels of interest the values are always greater than 0.7 over North Africa (Figure 2). There is a much wider range of emissivity values in the 8.7 $\mu$m channel than in the 10.8 and 12.0 $\mu$m channels (panels (a) to (c)): for the latter two channels, the minimum values are greater than 0.9. A characteristic reduction in the emissivity occurs within the approximate range of 8-10 $\mu$m, due to the reststrahlen absorption band in quartz silicates (e.g. Wald et al., 1998; Seemann et al., 2008), and hence sandy soils tend to have a much lower emissivity at 8.7 $\mu$m than do rockier or more vegetated surfaces. As a result, the geographical and even the temporal variability in the 8.7 $\mu$m emissivity will be a strong influence on the variability of the pristine-sky green values in the SEVIRI dust composite imagery. In this study we make use of surface emissivity values derived by Borbas and Ruston (2010) using MODIS data, following techniques developed by Seemann et al. (2008).

Figure 2(d) subdivides the Saharan domain within three ranges of the surface emissivity at 8.7 $\mu$m, betweeen 0.7-0.8, 0.8-0.9, and 0.9-1.0. There are differences in the climatological values for June and July, accounting for those grid cells which belong in two different zones ('low-medium' and 'medium-high') at different times. 'Low' emissivity zones generally correspond to dune fields and sand seas, 'medium' zones to rockier desert hamadas and regs, while the 'high' emissivity zones encompass most other surface types including mountains and vegetated areas. It is not generally possible to distinguish between mountains and vegetated areas using the surface emissivity alone. To summarise this, Table 2 quantifies the percentages of the total data subset within each emissivity zone: $\sim$41% lie in each of the low and medium emissivity zones. 18% lie in the high emissivity zone, a consequence of the strict cloud-screening criteria implemented in subsampling the output model data, with clouds more frequently occurring over highly emissive vegetated surfaces.

There is evidence from laboratory measurements that the thermal emissivity in the 8.2-9.2 $\mu$m range can increase by up to 16% with increased surface moisture (Mira et al., 2007), with respect to the dry soil case, especially for sandy soils. Li et al. (2012) argue that this results in a diurnal variability in emissivity, with moister soils and hence higher emissivity values at night. Within the seasonal context, it is therefore to be expected that Sahelian and southern Saharan soils in particular will be moister with the onset of the monsoon in late June. This is very apparent for example in northern and eastern Mali between June and July: in the central Sahara in northern Mali many points flip from being in the low to being in the medium emissivity zones, with emissivity values hovering around 0.8 (Figure 2(d)).

Considering a priori the overall effect of the surface emissivity on the pristine-sky imagery colours and neglecting the differences in atmospheric transmission, it is apparent that the slightly higher emissivity values at 12.0 $\mu$m than at 10.8 $\mu$m would give rise to very slightly positive brightness temperature differences (BTDs) between these channels and hence moderately high pristine-sky red values (Equation 1). Over low 8.7 $\mu$m emissivity surfaces the 10.8 $\mu$m emissivity values will be much higher, as will be the $T_B$ values, giving rise to high green values. For high 8.7 $\mu$m emissivity surfaces the contrast between the channels is much reduced, and so the green values will be correspondingly smaller. The combination of these is presented in



Figure 2(e), which presents the surface-only contribution to the Desert Dust RGB composite imagery, calculated from these emissivity values and assuming a fixed skin temperature of 30 °C. There is thus a high degree of geographical diversity in the colours arising purely from the intrinsic surface properties.

### 3.3 Skin temperature and column moisture

In the infrared the primary control on the measured radiances is temperature. The radiative emission temperature at the land surface is referred to as the skin temperature ($T_{\mathrm{skin}}$). High skin temperatures are characteristic of central desert regions during the day, while the coolest skin temperatures occur most frequently on the desert margins and at night. The desert skin temperature has a strong amplitude in its diurnal cycle. Analogously to the emissivities, we define five $T_{\mathrm{skin}}$ ranges with widths of 14 K, encompassing the total range of possible values in this data subset (Table 2). 'Cold' temperatures of $< 286$ K occur mostly at

night and are comparatively rare, as are 'very hot' temperatures of $> 328$ K, which occur only during the day. Specifying the daytime and night-time distributions more precisely, for the five $T_{\mathrm{skin}}$ ranges the daytime (0900, 1200, and 1500 UTC) distributions are (0.0001, 1.73, 38.29, 56.79, 3.19)%; at night-time (0000, 0300, and 2100 UTC) the distributions are (0.34, 52.48, 47.18, 0.01, 0.00)%. Meanwhile the 0600 and 1800 UTC timeslots display mixed day/night conditions. Across the entire diurnal cycle the 'warm' (300-314 K) $T_{\mathrm{skin}}$ range contains the most points (48%), as a common temperature range for both day and

night.

However, the influence of the skin temperature on the composite imagery colour can be masked by the variability in atmospheric water vapour (also referred to here as the total column water vapour, TCWV) and the consequent inter-channel differences in the atmospheric transmittance. Water vapour is a major absorber of IR radiation, and the differences in the channels' sensitivity to its presence govern the colour response of the imagery to variable water vapour content. As is to be expected,

the simulated transmittances across all three window channels are strongly negatively correlated with increasing water vapour. It has been noted before that the 12.0 µm channel has greater water vapour continuum absorption than either the 8.7 or the 10.8 µm channels, leading to lower atmospheric transmittance values in this channel (e.g. Brindley and Allan, 2003). This is confirmed by Table 3, which presents the mean COSMO-MUSCAT-RTTOV calculated channel transmittances, within the four column moisture ranges defined in Table 2. The transmittance is at its maximum in dry atmospheres in the 10.8 µm channel,

while it is most reduced under wet conditions for the 12.0 µm channel.

The geographical relationship between the average column moisture and the average channel atmospheric transmittances as simulated by COSMO-MUSCAT-RTTOV is mapped in Figure 3. The ITD is apparent in a winding belt between $\sim 18$ and 21°N (panel (a)), south of which are the moister monsoonal atmospheres of southern West Africa, and north of which are the characteristically drier atmospheres of the central Sahara. Considering the overall distribution of column moistures (Table 2),

points are relatively infrequent in the moister regimes, compared to the percentage of points in the two drier regimes: 83% of points have a TCWV of less than 26 mm. The geographical contrast in the transmittances is especially apparent in the 12.0 µm channel (panel (d)), with very low values south of the ITD. The weakest response to water vapour is apparent in the 8.7 µm channel. Meanwhile it is clear that the highest transmittances occur in 'dry' and 'dry-moist' atmospheres in the 10.8 µm channel, which are the conditions most frequently encountered in the region. Hence the 10.8 µm brightness temperatures will



have the greatest sensitivity to changes in skin temperature, while the brightness temperatures in the $12.0\,\mu$m channel will be the most responsive to changes in the column moisture, as has been noted before (Brindley et al., 2012; Banks et al., 2018).

### 3.4 Dust AOD and altitude

As a proxy for the quantity of dust in an atmospheric column, the AOD is an obvious property related to the colour response of the false-colour IR imagery in the presence of dust. It is calculated by COSMO-MUSCAT at a wavelength of $550\,$nm. B2018 showed that employing the VO73 dust refractive indices with typical COSMO-MUSCAT simulated dust size distributions leads to a peak in spectral extinction at $10.8\,\mu$m. For a given dust AOD the brightness temperatures at $10.8\,\mu$m will therefore be reduced more than those at 12.0 and $8.7\,\mu$m, behaviour also noted by Brindley et al. (2012) and exploited by the original Desert Dust imagery formulation, since following Equations 1 and 2 deeper red and weaker green colours occur with increasing AOD, triggering characteristic pink colours. Mean simulated dust AODs are mapped in Figure 4. In summer there are particularly active dust source regions in the south and west of the Sahara, subject to convective systems protruding northwards from the wetter regions of southern West Africa. Moreover, dust has a tendency to linger within the SHL, and hence large areas of northern Mali and western Niger see mean simulated AODs of greater than 1 over the six months.

The atmospheric temperature of the dust layer has a close relationship with the altitude of the dust layer, and it is of particular relevance since the temperature contrast between the background surface and the dust layer can be used to discriminate dust in IR measurements (e.g. Legrand et al., 2001; Brindley, 2007). Higher dust tends to be colder and to have the greatest contrast with the surface, although this is not necessarily the case at night or in colder seasons. It is also to be expected that differential atmospheric absorption between channels would also be an influence on the colour of the composite imagery for particularly elevated dust with respect to the background surface. To explore this impact of height, it is useful first to specify a representative height of IR optically active dust in an atmospheric column, considering also that there may be more than one layer. Therefore considered here is an average height defined by the AOD at $10.8\,\mu$m: this is referred to as the dust-AOD median height, at which half of the AOD is below this height. It is defined above ground level, on the basis that it is the contrast between the atmospheric and the surface temperatures which is of most significance for the colour sensitivity.

Defined dust height ranges are specified in Table 2, with 'Layer 3' (2-3 km) being the layer with the most frequent dust heights, at 35%. In the case of the AODs, the lowest AOD regime ('near-pristine', AOD $< 0.2$) has the highest frequency of points (44%), with higher AOD values becoming progressively less common. Over the whole dataset there is an anti-correlation between the AOD and the dust height, but it is weak (-0.30), an indicator of the diversity of the simulated dust storm situations.

### 3.5 Co-location of surface and atmospheric properties

Given the multitude of factors influencing the colour of dust in the imagery, a first step in disentangling their interlocking influences is to quantify the extent to which each of the properties co-vary with each other. Behaviours which may appear to be correlated with one variable may in fact be caused by another variable co-located with it. One may expect for example, the sandy deserts characteristic of the low emissivity zone to be characterised also by particularly hot skin temperatures, dry atmospheres, and by high dust AODs. Meanwhile as noted before by, e.g., B2018, it can be difficult to perceive dust in the





imagery over high emissivity surfaces (which may be more vegetated, for example), but is this due to the surface or due to a co-varying moist atmosphere?

Figure 5 plots the co-location of the variables with reference to the AOD regimes, the key metric identifying the intensity of dust activity. The principle of this plot is to identify the distributions of the variables within the AOD regimes: within each

plotted regime, the total is 100%. Panel (a) considers how the emissivity zones are distributed within the AOD regime subsets: at the lowest AODs the distribution of emissivities is quite close to the overall 41/41/18% split in the emissivity zones (Table 2), indicating no preferred zone for pristine conditions. However as the AOD increases, the dust events become increasingly weighted away from the high emissivity zones and towards instead the medium emissivity zone. As indicated by Figures 2(d) and 4, many of the high AOD areas are in eastern Mali, southern Algeria, and western Niger, in the medium emissivity zone.

Figure 5(b) considers the distributions of the $T_{skin}$ ranges within the AOD ranges. The 'warm' (300-314 K) range is always the most frequent, sitting as it does between the daytime and night-time distributions. High AODs are not associated with cool skin temperatures, but they are commonly co-located with warm temperatures. Hot skin temperatures appear to vary little in their proportions with AOD, while cool temperatures are more of a feature of light-dust environments. In terms of moisture (panel (c)), near-pristine scenes are particularly dry, and it is apparent for this June-July period that dust storms are not characteristic of

dry atmospheres within the simulations, rather they are accompanied more typically by moister atmospheres. This association of dust events with increased moisture has been observed before, by, e.g. Marsham et al. (2016). The mean column moisture within the near-pristine AOD (0.0-0.2) regime is 14.7 mm, while in the thick dust AOD (2-3) regime the mean is 28.4 mm. Referring back to Section 3.1 and to Figure 4, it appears that intrusions of moist monsoonal air from the south and the presence of the SHL are the dominant features of the major dust storm patterns simulated by COSMO-MUSCAT during this period.

The optically thickest dust storms (Figure 5(d)) tend to have defined heights within the bottom 2 km of the atmosphere: the proportion of dust in the lowest two height layers increases markedly with increasing AOD. Meanwhile dust only tends to be calculated to be within the higher altitude ranges when the simulated dust loading is light. Knowledge of the water vapour vertical distribution is also of significance for understanding how the dust signal at TOA may be obscured by water vapour (WV): defined here is a representative water vapour height above which the atmosphere is 'dry' (i.e. < 13 mm), in support

of the analysis in Section 4.2, since it is the quantity of water vapour above the dust layer which is of most interest for its influence on the dust signal. Near-pristine scenes are simulated to be relatively dry (panel (b)) and hence the WV heights tend to be correspondingly low (< 1 km). Thicker dust is often simulated to be accompanied by increased moisture content, and so the WV heights become more elevated, especially common within the 1-2 km layer. In comparison with panel (d) therefore, it is very often the case that the dust layer is suspended within a similar height range as the water vapour, which will have

consequences for the apparent colour of dust in the synthetic imagery.





## 4 Consequences of surface and atmospheric properties for the colour of dust in Desert Dust imagery

### 4.1 The night-time challenge to the perception of the colour of dust in the Desert Dust imagery

The night-time period poses its own challenges for the Desert Dust imagery. Infrared observations show different characteristics compared to during the day, due to the different contrasts in temperature between the surface and the atmospheric dust layers.

This is a known challenge for automated dust detection (e.g. Ashpole and Washington, 2012). An example of how difficult it can be to perceive dust at night even with IR imagery is apparent in Figure 1(b), and is also presented by the synthetic imagery in Figure 6. The same gradually evolving dust storm, 12 hours apart and with AODs of up to 3-4, appears very different in the synthetic imagery between daytime and night-time. The colour values of the dust itself are not in fact that much different between the day and night cases: for AODs > 3 the mean red and green values are 0.84 and 0.56 at 1200 UTC (from 148 grid

cells), and 0.85 and 0.55 at 0000 UTC (141 grid cells). However what is different is the colour of the surrounding background environment, which is often light blue during the day but is a murky purple during the night. The contrast between the dust and the background environment is hence typically much clearer during the day than during the night, a result of the strong diurnal cycle in skin temperature.

  The overall consequence of the surface cooling during the night on the simulated colours is displayed in Figure 6(e, f),

updated from Figure 10 by B2018. The panels display the mean colours, with respect to the red and green beams, as a function of the AOD and emissivity regimes (Table 2) for day and night conditions. The progression of the mean colours with increasing AOD for each emissivity regime is marked by the black lines. So for example in panel (e), over sandy desert surfaces ($0.7 < \epsilon < 0.8$, marked by triangles) during the daytime, near-pristine conditions (red symbols) are marked by medium red colour values and high green values, resulting in a light blue background colour. As the AOD increases, the red values

are boosted and the green values are suppressed, leading to pinker dust colours in the composite imagery at the highest AODs (black symbols). The differences between channel brightness temperatures are critical, and explain the apparent colour patterns. The diurnal cycle in near-pristine $T_{B108}$ values is greater than the cycle in the other two channels, hence $T_{B108}$ can be greater than $T_{B120}$ during the day but lower than $T_{B120}$ at night, giving rise to redder colours at night. This is a simple consequence of the atmospheric transmittances at $10.8\,\mu$m being greater than the transmittances in the other two channels under all but the

wettest atmospheres (Section 3.3, Table 3), such that $T_{B108}$ is more sensitive to the skin temperature than the other channels.

  The diurnal cycle in the near-pristine brightness temperatures (not shown) has a greater amplitude than does the diurnal cycle in the 'thick' dusty brightness temperatures, such that on average the presence of dust during the day tends to cool the brightness temperatures with respect to the pristine case. During the night the opposite occurs, when the presence of dust acts to warm the brightness temperatures compared to the near-pristine case, something observed before by other authors (e.g.

Legrand et al., 1988). In the simulations, 'thick' dust layers with AODs between 2 and 3 are most prevalent in the bottom 2 km of the atmosphere (Figure 5(d)), in which height range the night-time air temperatures are not much cooler than the skin temperature, and can often be slightly warmer. With a weak diurnal cycle in the air temperature the diurnal cycle in the thick dust brightness temperatures is correspondingly weaker than the diurnal cycle in the near-pristine brightness temperatures. This is a recognised problem for dust AOD retrievals using IR channels (e.g. Brindley and Russell, 2009; Banks and Brindley,





2013) which quantify the presence of dust using the assumption that the dust layer is cooler than the background surface, and which in consequence do not attempt to produce retrievals at night.

For thicker dust loadings, the colour patterns show much more similarity between day and night. The differences between the daytime and night-time $T_{B120}$-$T_{B108}$ differences are small, such that the resultant differences in thick dust mean red colours
between day and night are negligible: amongst the emissivity zones the mean red colours in the thick dust case (AODs between 2 and 3) are between 0.70 and 0.73 during the day, and between 0.71 and 0.72 at night. The $T_{B108}$-$T_{B087}$ difference and the resultant green colours tell a similar story in the thick dust case. Given a sufficient quantity, dust appears very much the same in the composite imagery throughout the diurnal cycle.

As a result of this, the lengths of the colour tracks in Figure 6(e, f)) are much shorter at night than during the day, and so
from a measurement perspective it is harder to resolve dust in the composite imagery at night. Longer colour tracks, as seen in panel (e), give greater clarity between dust and non-dust scenes. It is still possible to perceive dust in the night-time imagery, especially when analysing consecutive images and thereby observing the subtle patterns of motion above the static background surface, but the overall clarity is reduced.

## 4.2 Dust colours simulated by COSMO-MUSCAT-RTTOV

Generalising the analysis introduced in Section 4.1, Figure 7 introduces the mean COSMO-MUSCAT-RTTOV simulated colours categorised for all combinations of the three emissivity zones with the cool to very hot skin temperature regimes, and with the four moisture regimes, in an attempt to disentangle the influences of the various environmental factors and to provide a physical interpretation of the imagery colours. The rings represent the colours within specified AOD and height ranges: the innermost ring is the 'near-pristine' case, with AODs between 0.0 and 0.2 over all height ranges; the outer four
rings display the output colours within the AOD range of 2-3 ('thick dust'), from the inner to the outermost rings subdivided by the height ranges of 0-1, 1-2, 2-3, and 3-4 km. The entire 3-hourly diurnal cycle is represented, the distinction between day and night being implicit only in the skin temperature regimes. Within the thick dust AOD range there are no points within the 'cold' skin temperature regime, and so for clarity these are not represented here. The numbers for each segment are the mean red values, which may be considered as the most vital of the constituent colours for producing characteristic pink dust colours:
with reference to Figure 6(e, f), pink colours become apparent between red values of $\sim 0.6$ and 1, and between green values of 0 and $\sim 0.8$. For the perception of dust it is more important that the red value is high than that the green value is low.

Considering first the 'near-pristine' case, mean red values are at their greatest over the low emissivity zone, the coolest skin temperature regimes, and under the driest atmospheres ('low-cool-dry'), the regime which has the highest mean red value of 0.80. Meanwhile the lowest value of 0.00 occurs in both the medium and high emissivity zones, in the very hot and wet
regime. The hottest surfaces have the lowest red colours, and hence are characterised by varying shades of blue. Of these, the lowest emissivity surfaces are the most light blue, while darker blue colours belong in the higher emissivity regimes. Light blue corresponds to high green values (from lower $T_{B087}$ values), and dark blue to lower green values. Cooler surfaces have higher red values, as seen in Section 4.1, and hence display more hints of purple. These hints of purple in the cool regime are at their strongest in the high emissivity zone, when the green values are at their lowest and hence the colours are deeper into



the purple/pink colour domain. This description generalises the 'night-time effect', of redder background colours during the night, as being rather a 'cool skin temperature effect' which is applicable not just to the diurnal but also to the seasonal cycle.

Notable in the near-pristine case are the ranges of available mean red colours over the moisture sets depending on whether the skin temperature is cool or very hot. For example, in the low-cool set the maximum red value is 0.80 in the dry regime,

and the minimum is 0.48 in the wet regime. In this low-cool set there is therefore a range of 0.32 in the mean red values. In the medium- and high-cool sets, the corresponding red ranges are 0.34 and 0.29. However when the skin temperature is 'very hot', the corresponding red ranges are much larger: 0.52 (low emissivity, only from dry to moist), 0.59 (medium emissivity), and 0.53 (high). In near-pristine conditions, for a given amount of moisture, the atmosphere is less transparent in the $12.0\,\mu$m channel than in the $10.8\,\mu$m channel, and this difference between the channels increases with atmospheric moisture content.

The contrast is exacerbated when the surface skin temperature is hot because there is more surface emission in both channels to interact with the overlying atmosphere, given the relatively flat surface emissivity between the two channels. This means that the near-pristine $T_{B120}$-$T_{B108}$ difference, which is often negative, becomes more negative when the surface is 'hot' and the overlying atmosphere is 'wet', with the converse being true for 'cool' and 'dry' conditions.

Analysing the ranges from the perspective of the skin temperature, the dry sets have red ranges between 0.18 and 0.20

(depending on emissivity), while the wet $T_{\mathrm{skin}}$ sets have ranges between 0.41 (cool to hot only) and 0.44. The near-pristine red colour appears to be more sensitive to variations in the column moisture than to variations in the skin temperature, an obvious caveat to which is that the skin temperature informs the behaviour of the colour response to moisture. Meanwhile the red values are weakly sensitive to the emissivity zones, with the red ranges of the emissivity sets varying between just 0.025 and 0.121 amongst the 16 $T_{\mathrm{skin}}$/moisture combinations.

In contrast, the green colours are more strongly governed by the emissivity, hence the distinctly different shadings between the emissivity zones. The green ranges of the emissivity sets vary between 0.17 and 0.30 (16 possible combinations), for the $T_{\mathrm{skin}}$ sets the ranges are 0.07 to 0.15 (12), and for the moisture sets the ranges are 0.00 to 0.23 (12). There is less variability in the green behaviour due to $T_{\mathrm{skin}}$ or moisture compared to that of the red, since the sensitivity of the transmittance in the $8.7\,\mu$m channel to water vapour is less than that of the $12.0\,\mu$m channel. The green beam is more a function of the surface properties

alone.

Introducing dust into the analysis, generally the 'near-pristine' colours tend to appear bluer than the 'thick dust' colours, which are redder and hence pinker. This is always the case in the hot and very-hot regimes, with the most exceptions to this pattern occurring in the cool regime for dust in lower altitude ranges (inner dust rings). For thick dust in variably moist atmospheres, it is the case in almost all circumstances that dry dusty atmospheres are the reddest and hence pinkest within their

respective emissivity-$T_{\mathrm{skin}}$-height regime. The corollary of this is that red/pink signals for a given amount of dust are at their weakest in the wettest atmospheres, confirming that moisture acts to 'hide' the presence of dust in the IR imagery. Meanwhile the simulated red colours of thick dust storms are broadly independent of the surface emissivity.

It is always the case that higher altitude dust regimes (outermost rings in Figure 7) are redder than lower altitude dust, giving rise to pinker colours. The greatest mean red values of 1.0 can reliably be found for dry, high-altitude dust, a scenario

which is insensitive to the emissivity or the skin temperature. This implies that higher altitude dust has higher values of $T_{B120}$



than of $T_{B108}$, compared to dust at lower levels. This difference can be explained by the optical path length between the dust layer and the TOA. Assuming the dust layer is optically thick and hence acts as the emitting 'surface', the optical path length (predominantly driven by the water vapour distribution) will be shorter for higher altitude dust, such that the impact of the variation in the 10.8 and 12.0 $\mu$m channel transmittances will be reduced (Table 4). The atmosphere is very regularly 'dry'

above dust in Layer 4 (3-4 km, as defined in Table 2), and is often drier than total atmospheric columns defined as 'dry'. Hence examining the segments in Figure 7 radially outwards, it is always the case that the red values increase towards the outermost ring (highest dust altitudes); similarly by examining each moisture set clockwise from wet to dry, it is almost always the case that the red values increase in the dry direction. Consider dust in the medium-warm-wet regime at an altitude of 0-1 km: it has a red value of 0.52 and would be very hard to discriminate as dust in a SEVIRI image. If this atmosphere is replaced with a dry

atmosphere then the red value is nudged up to 0.74, making it slightly easier to identify it as dust, but not convincingly. If instead the dust in the wet atmosphere is moved up to between 3 and 4 km, then the red value is boosted to 0.81 and characteristic pink dust colours become apparent. Especially in the warm and cool regimes, dust in wet conditions may be clearly observed if it is at a sufficiently high altitude.

It is clear that dust height has a substantial impact on the apparent colour of dust in the IR imagery. If dust is in the bottom

layer of the atmosphere then it is only likely to be detectable in the imagery if the skin temperature is particularly hot and the atmospheric column is particularly dry. However as shown by Figure 5(c) it is rarely the case in the simulations that the column moisture is 'dry' within this AOD range (3.0 %), and so low-altitude thick dust will regularly be missed in the imagery. High altitude thick dust ($> 3$ km) is always likely to be visible in the imagery, given the strong colour contrast between it and the background surface. Even under the wettest atmospheres high dust will be clearly noticeable, since despite possible purple

dust colours there is a strong colour contrast with the dark blue background surface which will be clearly distinguishable in successive imagery. However Figure 5(d) indicates that thick dust rarely has a defined height as high as 3-4 km.

Figure 7 has provided a physical interpretation of the consequences of different environmental regimes for the apparent dust colour. It is worth considering how frequently such conditions occur, in order to identify those segments within Figure 7 which represent the most likely scenarios. Within the six-month June and July simulation period the most common height range for

thick dust is calculated to be in Layer 2, between 1 and 2 km (57 %). Such dust is unlikely to be noticeable above cool surfaces, although within this AOD and height range the warm skin temperature regime is the most common (71 %). In winter it is to be expected that the percentages would be weighted more towards the cooler regimes.

Subdividing further within the regime of thick dust at 1-2 km height, Table 5 shows that the medium emissivity, warm skin temperature, and moist column water vapour regime is the combination simulated to occur most frequently (26 % of the 48

possible combinations) during this period. Identifying this segment in Figure 7, it is apparent that this dust has a red colour of 0.70, displaying pink-purple colours, with a noticeable but not substantial contrast against the blue background surface, which has a red colour of 0.44. Table 5 indicates that most frequently attention should be focused on the dry-moist, moist, warm, and hot regimes. Of these, dust in the hot and moist regimes displays the most distinct simulated colour contrasts with the background surface. The more frequent warm points display murkier contrasts against the background surface, although the

dust is still likely to be readily apparent in the imagery.





B2018 showed that the simulated colours produced by COSMO-MUSCAT-RTTOV are likely to be insufficiently deep, when compared with SEVIRI observations and retrieved AODs. It is plausible that the near-pristine colours should be greener and less red, implying that the COSMO-MUSCAT simulated skin temperatures are likely to be too cool; meanwhile the thick dust colours should perhaps be redder and less green, implying that in the simulations the dust extinction properties are too weak, or that the atmosphere is simulated to be too moist. An implication of this is that dust in the real imagery may be visible at lower AODs, and/or that it may be visible at lower altitudes than in the synthetic imagery.

In summation, cool skin temperatures boost the red beam when the dust loading is light, while hotter skin temperatures suppress it, a result of the high atmospheric transmittance at $10.8\,\mu m$. Moister atmospheres will display weaker red colours, a result of the higher atmospheric absorption at $12.0\,\mu m$ due to water vapour compared to the other channels. Thicker dust loadings boost the red beam and reduce the green beam, a result of the greater IR absorption by dust at $10.8\,\mu m$. Dust at higher altitudes has a shorter atmospheric column above it, and hence is less masked by water vapour, giving rise to redder and more vivid pink dust colours in the IR imagery.

## 5 Conclusions

This paper is the follow-up paper to Banks et al. (2018), which explored the sensitivity of the colour of dust in SEVIRI Desert Dust IR imagery to dust optical properties. Using the COSMO-MUSCAT-RTTOV modelling system, this paper has explored the sensitivity of the colour of dust in SEVIRI Desert Dust IR composite imagery to various environmental properties, including the surface thermal emissivity, surface skin temperature, atmospheric column moisture and atmospheric transmittance, dust AOD, and dust height. These properties are often co-varying with each other. The relationships between all these variables are intricate, but display distinct patterns in the colour of the imagery, patterns which can be discriminated by considering numerous combinations of the environmental variables.

The surface thermal emissivity (at $8.7\,\mu m$) is a controlling variable on the colour, in particular the green beam, of the near-pristine imagery. It is less significant at higher AODs, although it still contributes to the intensity of the green beam. Skin temperatures are also an important factor governing the colour of near-pristine imagery, with hotter skin temperatures tending to give rise to bluer (i.e. less red) colours than cooler surfaces. Cooler surfaces are redder, explaining why night-time and winter Desert Dust imagery is characterised by a greater prevalence of pink and purple hues across the desert surface, while hot summer and daytime surfaces are characterised by light blue colours. Pink atmospheric dust is much easier to discriminate against the light blue summer daytime surfaces.

The significance of the skin temperature for the colour of the imagery is related to the atmospheric water vapour content: water vapour is more absorbing at $12.0\,\mu m$ than at $10.8\,\mu m$ and hence the wetter the atmosphere the weaker the red colour. The atmosphere is more transmissive at $10.8\,\mu m$ and so the brightness temperature in this channel is more sensitive to the skin temperature. A stronger diurnal cycle in $T_{B108}$ than in $T_{B120}$ gives rise to a diurnal cycle in the red values whereby the red values are greatest at night. Similarly, wet atmospheres also act to hide the presence of dust by reducing the apparent red colours and hence the distinctive 'pink dust' colour.





The importance of the dust height in producing redder and pinker colours is clearly shown by Figure 7 (understanding colour with reference to dust height is a novel feature enabled by the use of the COSMO-MUSCAT-RTTOV system), but it may be an easy misconception to make that this is due to the larger temperature contrast between elevated dust and the background surface. Of more relevance is the difference in atmospheric transmittance above the dust layer between the 10.8 and 12.0 $\mu$m

channels, the relative difference between which reduces for higher atmospheric dust layers. Some analogy can be made between high dust and dry dust, in that the atmospheric absorption at 12.0 $\mu$m due to water vapour is reduced when the dust is elevated higher. Hence the most reliable red and pink colours arise from high-altitude dust in dry atmospheres. Meanwhile low altitude dust can often be hard to discriminate from the background surface, but this is not because the dust layer is warm, but instead because the dust is more readily hidden by atmospheric water vapour the lower in the atmosphere it is located.

In the Desert Dust RGB composite imagery, however thick the dust loading, dust is very likely to be invisible when it is very low in the atmosphere ($< 1$ km), in an altitude range where it is very likely to be obscured by water vapour. In contrast, dust elevated to altitudes greater than 3 km will always be likely to be apparent in the imagery, and will have a lower AOD threshold for perception. It is usually difficult to distinguish dust within particularly wet atmospheres, but if the dust is high enough then the atmosphere above it will be drier and hence the dust itself will become more readily apparent.

A possible extension of the insights provided by this paper would be to refine further and optimise dynamically the Desert Dust RGB composite imagery in order to highlight better the presence of atmospheric dust, if the background environmental properties are known or can be estimated. An example might be to reduce the range of the brightness temperature differences in the red beam when the skin temperatures are cool, requiring a mathematical relationship between the skin temperature and the red range to be determined. Such a technique could be performed with the aid of model simulations, or in future with the

aid of atmospheric sounding information from the Infrared Sounders that will be included onboard the upcoming Meteosat Third Generation (MTG) series of satellites. It is to be expected that the MTG satellites will have enhanced capabilities in the quantification of dust loading and properties, and there is potential for the COSMO-MUSCAT-RTTOV modelling system to be used to test the feasibility of future dust quantification techniques.

    In this paper we have presented a thorough evaluation of the various environmental factors that can influence the appearance

of the SEVIRI Desert Dust imagery product as simulated by the COSMO-MUSCAT-RTTOV modelling system. The results should be taken in conjunction with those of our previous paper (B2018) which considered the role of variability in dust optical properties alone. We hope that users of the imagery will find both analyses helpful in terms of assessing the strengths and weaknesses of the imagery as a dust identification and tracking tool. In particular, we also believe that users should be aware of the environmental properties which affect the appearance of the dust imagery, to avoid erroneously attributing these effects

to dust properties.

*Data availability.* COSMO-MUSCAT and RTTOV model output data are available on request from the authors. SEVIRI data are available from EUMETSAT, currently at https://www.eumetsat.int/website/home/Data/index.html. The RTTOV program is available from EUMET-SAT's NWP SAF facility, currently at https://nwpsaf.eu/site/software/rttov/.



*Author contributions.* JRB performed the RTTOV simulations and the analysis, and designed and wrote the manuscript. AH provided access to the SEVIRI data and discussion on the satellite remote sensing aspects, BH provided support for the COSMO-MUSCAT simulations and advice from the modelling perspective, HEB contributed to discussions on the interpretation of the dust imagery, HD contributed to ongoing discussions on the project design, and KS provided advice and support throughout the process.

5 *Competing interests.* The authors declare that they have no conflict of interest.

*Acknowledgements.* Jamie Banks and Kerstin Schepanski acknowledge funding through the Leibniz Association for the project "Dust at the Interface – modelling and remote sensing" (grant number SAW-2015-TROPOS-4). Deutscher Wetterdienst (DWD) have provided access to the COSMO model, as well as boundary data. We thank also EUMETSAT for producing the SEVIRI data, which have been obtained from the TROPOS satellite data archive.



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


**Table 1.** Size ranges, effective radii ($\mu$m), extinction efficiencies at 550 nm ($Q_{\text{ext,550 nm}}$), and IR extinction efficiencies of the five COSMO-MUSCAT size bins. Extinction efficiencies at 550 nm are derived for spherical particles from refractive indices developed by Sinyuk et al. (2003). IR extinction efficiencies ($Q_{\text{ext,ch}}$) are calculated for spherical VO73 dust, convolved over the MSG-2 SEVIRI filter functions.

| Bin | Radius range | Effective radius | $Q_{\text{ext,550 nm}}$ | $Q_{\text{ext,087}}$ | $Q_{\text{ext,108}}$ | $Q_{\text{ext,120}}$ |
|---|---|---|---|---|---|---|
| 1 | 0.1 - 0.3 | 0.169 | 1.677 | 0.072 | 0.065 | 0.025 |
| 2 | 0.3 - 0.9 | 0.501 | 3.179 | 0.225 | 0.211 | 0.081 |
| 3 | 0.9 - 2.6 | 1.514 | 2.356 | 0.905 | 1.333 | 0.548 |
| 4 | 2.6 - 7.9 | 4.570 | 2.144 | 2.083 | 3.033 | 3.592 |
| 5 | 7.9 - 24.0 | 13.800 | 2.071 | 2.213 | 2.454 | 2.543 |

(a)

(b)

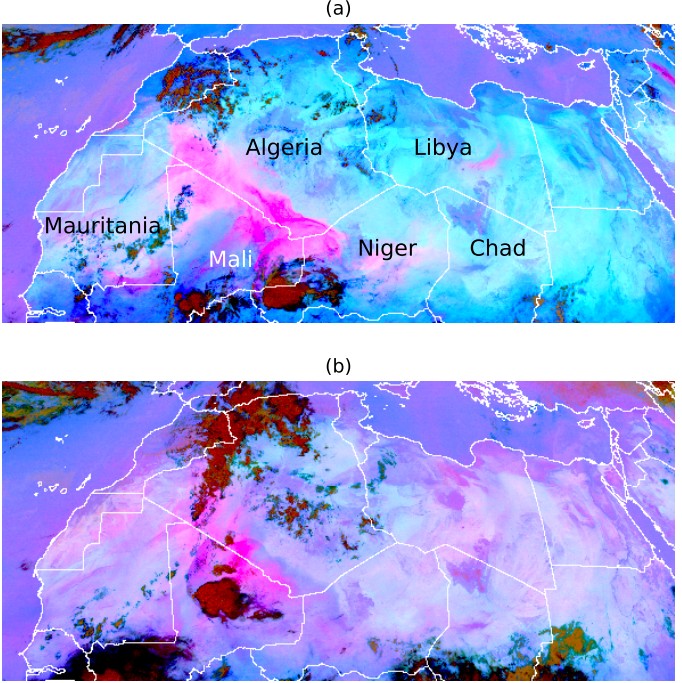

**Figure 1.** SEVIRI Desert Dust RGB images from (a) 1200 UTC on 17th June 2012, and (b) 0000 UTC on 18th June 2012.





**Table 2.** Distribution of variables within their specified ranges. $\epsilon$ is the emissivity at 8.7 $\mu$m, $T_{\text{skin}}$ is the skin temperature, and TCWV is the total column water vapour. The height refers to the dust height as specified by the median height of the AOD at 10.8 $\mu$m. Within each variable set, the sum is 100%. The total number of points is approximately 15.8 million, the maximum value of TCWV is 53.0 mm, the maximum simulated AOD is 14.8 (0.007% of points have an AOD greater than 5), and the maximum defined height is 19.7 km. The maximum AOD of points within Layer 6 is 0.024.

| Variable range | Description | Percentage (%) |
|---|---|---|
| $0.7 < \epsilon < 0.8$ | Low | 40.71 |
| $0.8 < \epsilon < 0.9$ | Medium | 41.31 |
| $0.9 < \epsilon < 1.0$ | High | 17.98 |
| $272 < T_{\text{skin}} < 286$ K | Cold | 0.19 |
| $286 < T_{\text{skin}} < 300$ K | Cool | 27.96 |
| $300 < T_{\text{skin}} < 314$ K | Warm | 47.77 |
| $314 < T_{\text{skin}} < 328$ K | Hot | 22.88 |
| $328 < T_{\text{skin}} < 342$ K | Very hot | 1.20 |
| $0 <$ TCWV $< 13$ mm | Dry | 27.96 |
| $13 <$ TCWV $< 26$ mm | Dry-moist | 54.90 |
| $26 <$ TCWV $< 39$ mm | Moist | 14.48 |
| $39$ mm $<$ TCWV | Wet | 2.66 |
| $0.0 <$ AOD $< 0.2$ | Near-pristine | 44.16 |
| $0.2 <$ AOD $< 0.5$ | Very light | 32.35 |
| $0.5 <$ AOD $< 1.0$ | Light | 16.40 |
| $1.0 <$ AOD $< 1.5$ | Light-medium | 4.65 |
| $1.5 <$ AOD $< 2.0$ | Medium-thick | 1.55 |
| $2.0 <$ AOD $< 3.0$ | Thick | 0.75 |
| $3.0 <$ AOD | Very thick | 0.14 |
| $0 <$ height $< 1$ km | Layer 1 | 5.27 |
| $1 <$ height $< 2$ km | Layer 2 | 23.04 |
| $2 <$ height $< 3$ km | Layer 3 | 35.33 |
| $3 <$ height $< 4$ km | Layer 4 | 26.66 |
| $4 <$ height $< 10$ km | Layer 5 | 9.68 |
| $10$ km $<$ height | Layer 6 | 0.02 |



**Table 3.** COSMO-MUSCAT-RTTOV simulated mean pristine-sky channel transmittances within the specified total column water vapour ranges ('dry', 'dry-moist', 'moist', and 'wet'), averaged over the distribution defined in Table 2, and marked with associated standard deviations.

| Moisture ranges (mm) | 8.7 $\mu$m | 10.8 $\mu$m | 12.0 $\mu$m |
|---|---|---|---|
| 0 < TCWV < 13 | 0.804 ± 0.026 | 0.882 ± 0.027 | 0.825 ± 0.040 |
| 13 < TCWV < 26 | 0.738 ± 0.036 | 0.806 ± 0.048 | 0.717 ± 0.064 |
| 26 < TCWV < 39 | 0.623 ± 0.038 | 0.638 ± 0.059 | 0.504 ± 0.070 |
| 39 < TCWV | 0.543 ± 0.022 | 0.515 ± 0.032 | 0.364 ± 0.034 |

**Table 4.** As with Table 3, COSMO-MUSCAT-RTTOV simulated mean atmosphere-only channel transmittances above the specified dust layers. These are averaged within the specified total column water vapour ranges (defined in Table 2 as 'dry', 'dry-moist', 'moist', and 'wet'), for points within the thick dust AOD range (from 2 to 3) for consistency with the outer rings of Figure 7. Included also are the mean water vapour column values above the specified dust layers.

| Moisture ranges (mm) | 8.7 $\mu$m | 10.8 $\mu$m | 12.0 $\mu$m | WV above dust (mm) |
|---|---|---|---|---|
| Layer 4, 3-4 km | | | | |
| 0 < TCWV < 13 | 0.911 | 0.966 | 0.949 | 3.2 |
| 13 < TCWV < 26 | 0.885 | 0.945 | 0.913 | 5.8 |
| 26 < TCWV < 39 | 0.880 | 0.940 | 0.905 | 6.7 |
| 39 < TCWV | 0.880 | 0.939 | 0.903 | 7.1 |
| Layer 1, 0-1 km | | | | |
| 0 < TCWV < 13 | 0.842 | 0.915 | 0.872 | 7.6 |
| 13 < TCWV < 26 | 0.784 | 0.852 | 0.780 | 14.4 |
| 26 < TCWV < 39 | 0.732 | 0.779 | 0.678 | 20.5 |
| 39 < TCWV | 0.662 | 0.676 | 0.546 | 28.2 |





**Table 5.** Percentages of simulated emissivity-$T_{skin}$-moisture occurrence within the thick dust AOD range (from 2 to 3), and within Layer 2 (1-2 km), ranges as specified in Table 2. The total of all values listed here is 100 %.

|  | Dry | Dry-moist | Moist | Wet |
|---|---|---|---|---|
| **Low emissivity** |  |  |  |  |
| Cool | 0.02 | 0.10 | 0.05 | 0 |
| Warm | 0.37 | 5.13 | 12.81 | 0 |
| Hot | 0.22 | 2.74 | 4.40 | 0.46 |
| Very hot | 0.001 | 0.04 | 0.003 | 0 |
| **Medium emissivity** |  |  |  |  |
| Cool | 0.12 | 0.48 | 0.05 | 0.003 |
| Warm | 0.90 | 14.12 | 26.13 | 3.95 |
| Hot | 0.28 | 6.27 | 9.67 | 1.51 |
| Very hot | 0.004 | 0.04 | 0 | 0 |
| **High emissivity** |  |  |  |  |
| Cool | 0.08 | 0.22 | 0.04 | 0.001 |
| Warm | 0.19 | 2.18 | 2.82 | 0.62 |
| Hot | 0.06 | 0.70 | 0.99 | 0.35 |
| Very hot | 0 | 0 | 0 | 0 |





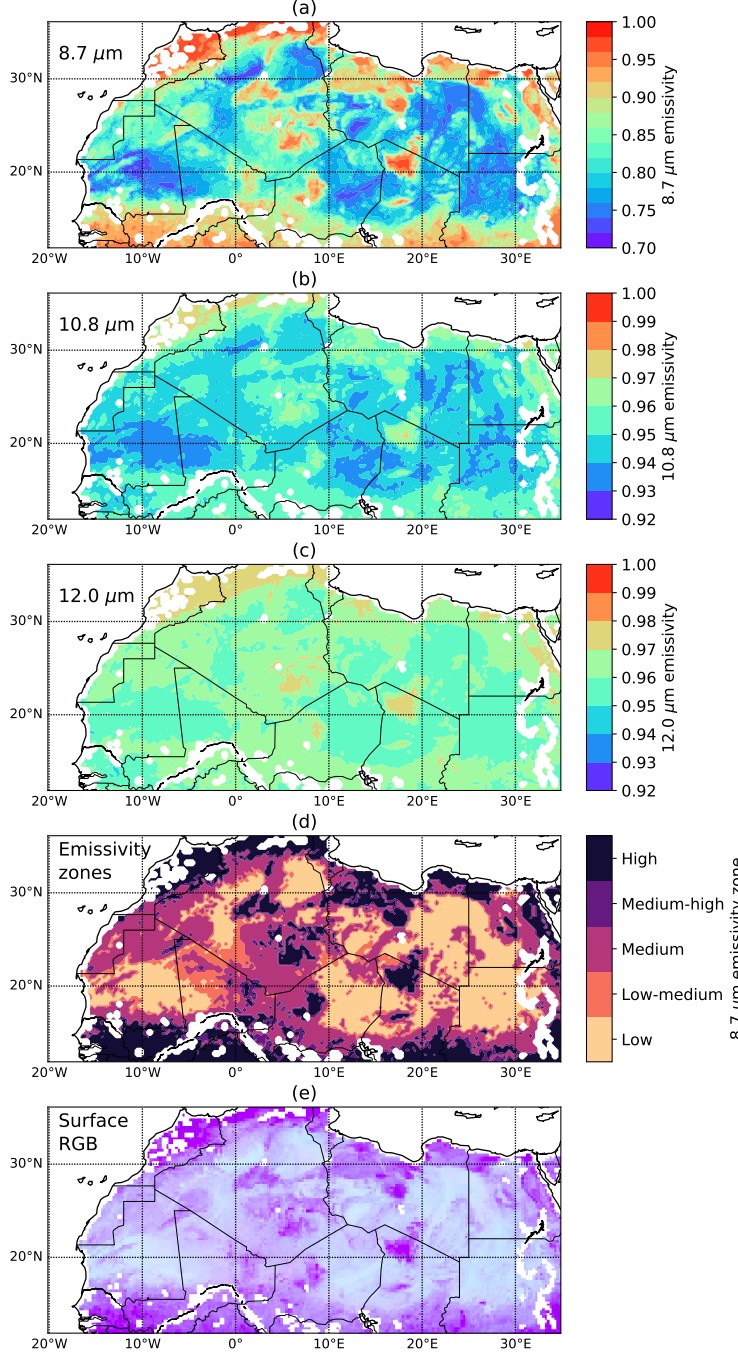

**Figure 2.** Maps of mean emissivities, Junes and Julys 2011-2013, for the SEVIRI channels at (a) 8.7 $\mu$m, (b) 10.8 $\mu$m, and (c) 12.0 $\mu$m. Note that panel (a) has a different colour scale to panels (b) and (c). These averages are built up from points where and when RTTOV simulations have been performed, i.e. only for land and cloud-free situations. Panel (d) specifies which emissivity zone each grid cell falls into, at 8.7 $\mu$m, which for some grid cells varies by month. The zones have emissivity ranges of 0.7-0.8 (low), 0.8-0.9 (medium), and 0.9-1 (high). Panel (e) depicts the Desert Dust composite image for the surface, calculated using the Planck function and using the emissivity values in panels (a-c) with a fixed skin temperature of 303.15 K.





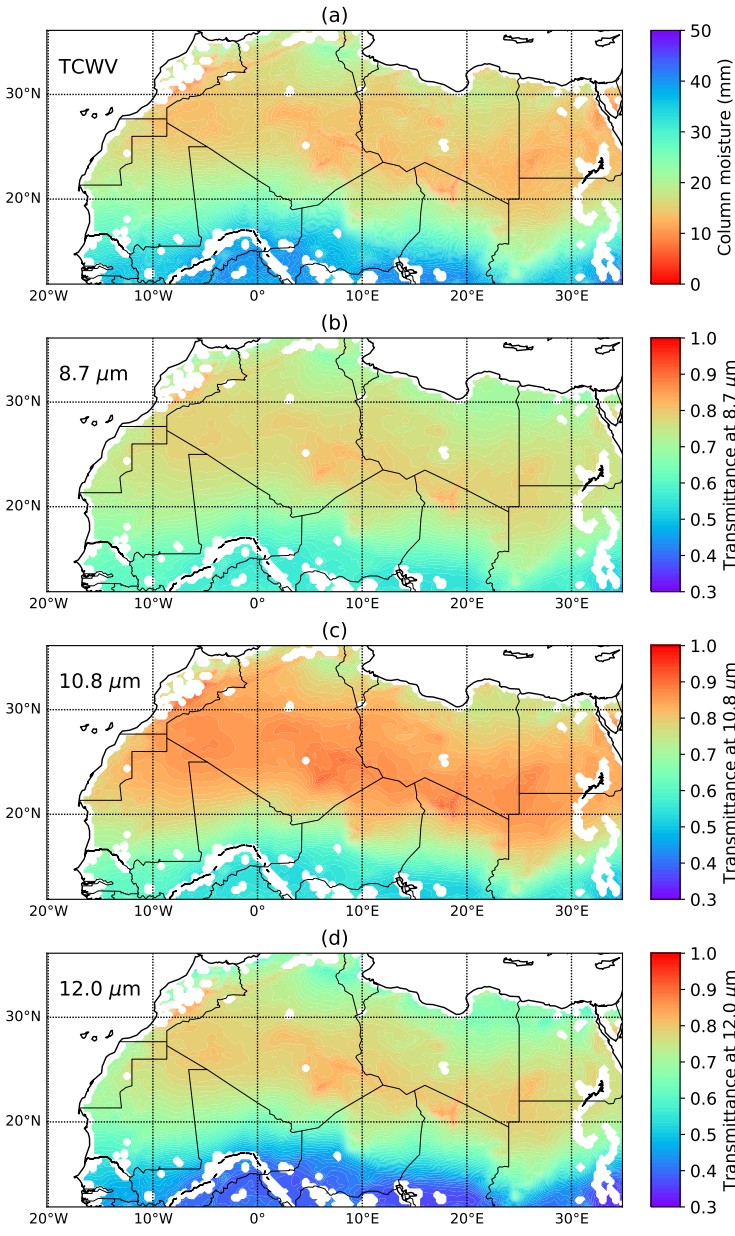

**Figure 3.** Maps of mean atmospheric properties simulated by COSMO-MUSCAT-RTTOV, Junes and Julys 2011-2013, for all eight timeslots during the diurnal cycle. (a) column moisture (mm); and (b, c, d) pristine-sky atmospheric transmittances at 8.7, 10.8, and 12.0 $\mu$m. As with Figure 2, these averages are only formed of points where and when RTTOV simulations have been performed.



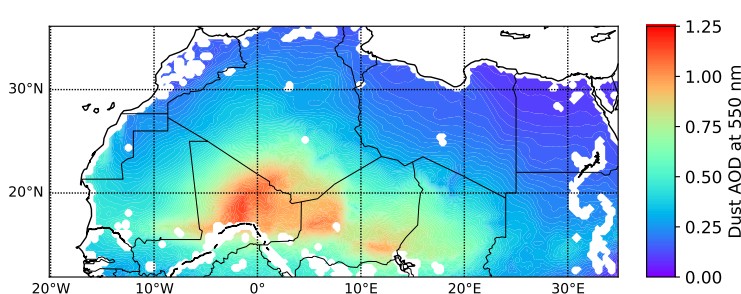

**Figure 4.** Map of mean dust AOD at 550 nm simulated by COSMO-MUSCAT, Junes and Julys 2011-2013, for all eight timeslots during the diurnal cycle. As with Figure 2, this average is only formed of points where and when RTTOV simulations have been performed.





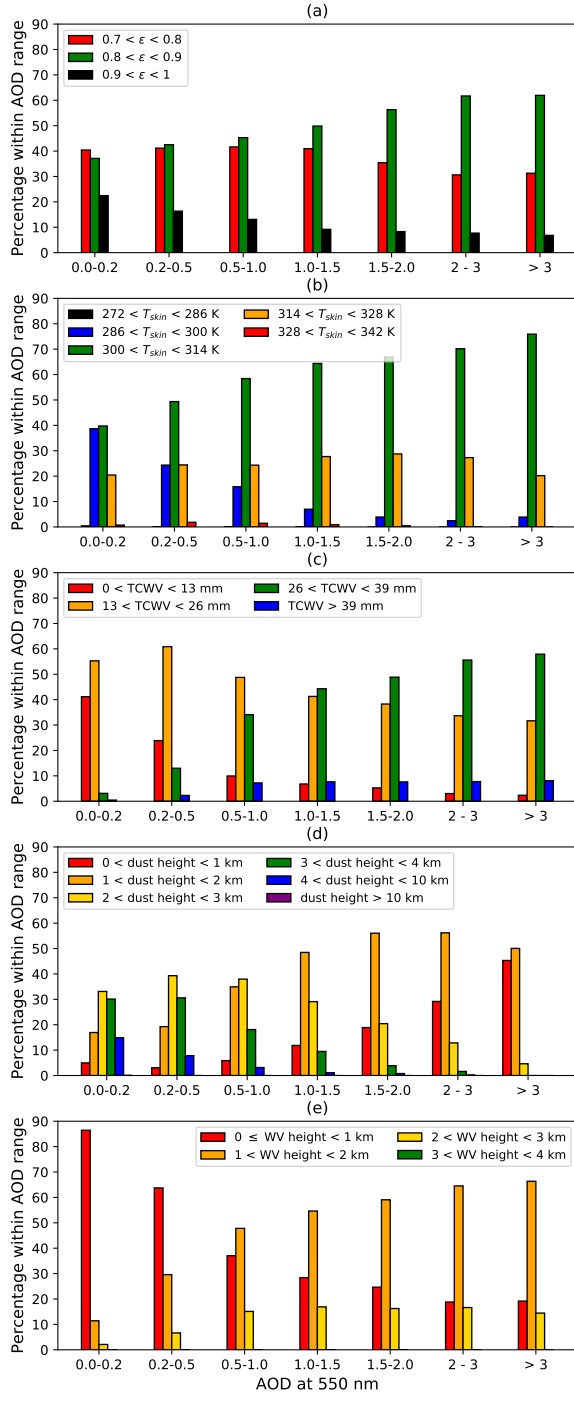

**Figure 5.** Bar charts indicating the percentages of occurrences of the subdivisions of the given properties, within ranges of the COSMO-MUSCAT AOD at 550 nm. The time period is for the Junes and Julys of 2011-2013, all 8 timeslots during the day. (a) emissivity at 8.7 $\mu$m; (b) skin temperature (K); (c) column moisture (mm); (d) dust height (km); (e) water vapour (WV) height (km), defined here as the height above which there is less than 13 mm of water vapour. Within each cluster of bars for a specified AOD range, the sum of the bars is 100%.





**Figure 6.** Maps of COSMO-MUSCAT simulated AODs (top row), and synthetic Desert Dust RGB images (middle row): (a, c) 1200 UTC on 17th June 2012; (b, d) 0000 UTC on 18th June 2012. In the bottom row are plots of mean colours with respect to COSMO-MUSCAT AODs (at 550 nm) and surface emissivity at 8.7 μm, Junes and Julys 2011-2013. Timeslots are 0900, 1200, and 1500 UTC (day) for panel (e), and 0000, 0300, and 2100 UTC (night) for panel (f). AOD ranges are marked by symbol colour as indicated, and emissivity ranges are marked by the symbol shapes, as indicated. The blue beam has a fixed value of 1. 0.04% of points have blue values of less than 1 during daylight hours, 5.47% at night. Error bars indicate the standard deviations associated with the colour means, but are only included for three of the AOD ranges, for clarity.





**Figure 7.** Mean colours simulated by COSMO-MUSCAT-RTTOV, subdivided into the ranges of emissivity (low-high), skin temperature (cool - very hot) and column moisture (dry-wet), as defined in Table 2. Values increase in the anti-clockwise direction, as indicated. The innermost ring is the 'near-pristine' case, with AODs < 0.2. The four remaining rings are the thick dust cases with AODs between 2 and 3, subdivided by height ranges. From the inner to the outer rings, the dust height ranges represented are 0-1 km, 1-2 km, 2-3 km, and 3-4 km (Layers 1-4). 0.2% of the points in this AOD range have a dust height of greater than 4 km, while no points in this range have a skin temperature of less than 286 K ('cold'). All 3-hourly timeslots from day and night are included. The numbers marked on each segment indicate the mean red value, to two decimal places, leading and ending zeroes removed for brevity. Black segments indicate coincident conditions not contained within this data subset.