# Peer review of "The sensitivity of the colour of dust in MSG-SEVIRI Desert Dust infrared composite imagery to surface and atmospheric conditions"

_Atmospheric Chemistry and Physics, 2018_

## Referee Comment (RC1) · Anonymous Referee #2 · 25 Jan 2019

This study examines under which conditions North-African desert dust can be distinguished in infrared RGB imagery from MSG-SEVIRI. The authors demonstrate that dust is in many cases not visible, in particular if the surface skin temperature is low, the atmosphere is moist or the dust is located close to the surface.

The analysis is based on a state-of-the-art modelling system and has been carried out in a very systematic way. The paper is clearly structured and excellently written. My only concern, apart from some minor specific comments listed below, is that the scope of the study should be better defined and motivated. The study is confined to northern African land, but Saharan dust is transported over the Atlantic Ocean and Europe.

What about the capabilities to detect dust in these regions, where the surface and atmospheric conditions (as well as the satellite viewing angles) are markedly different? The study is furthermore confined to two summer months. What about the rest of the year?

Specific comments:

Page 2, line 4-8: Can you mention a concrete example of a feedback loop?

Page 3, line 6-8: It is not clear to me what the significance of this statement is for the current paper.

Page 5, line 9: Is the solar zenith angle relevant here?

Page 5, line 14-15: I understand that you focus on cloud-free scenes, but it would be useful to say a few words about the signature of clouds in the imagery. And what about scenes with both clouds and dust?

Page 7, line 34: What does 'these' refer to?

Page 10, line 20-22: Can you explain why the thickest dust storms are located low in the atmosphere while lower loadings occur more frequently at higher elevations? Is it simply because the thickest loadings occur close to the (storm) source region and so have not had time to reach higher altitudes?

Page 16, line 21: The imager on MTG (FCI) will have somewhat different infrared channels than SEVIRI. For the three channels used in the desert dust RGB, the central wavelengths will move from 8.7, 10.8, and 12.0 micron to 8.7, 10.5 and 12.3 micron, respectively. Is there something to say about whether and how this will impact the detectability of dust?

Page 26: Could you add in the caption of Fig. 2 what the white patches correspond to. I assume there is no data because of nearby water surfaces, but there are also some white patches in the middle of the Sahara. Are these oases?

---

## Referee Comment (RC2) · Anonymous Referee #1 · 12 Feb 2019

Review

Title: "The sensitivity of the colour of dust in MSG-SEVIRI desert dust infrared composite imagery to surface and atmospheric conditions"

I found this work very interesting. I believe it is a notable contribution to the dust detection over the African desert using thermal infrared information from meteorological satellites. It is well written, the physical explanation of your findings is analytic and the main conclusions are very useful for the potential readers and the relative scientific community.

General comments 1. There is no relative information about the spatiotemporal correlation between COSMO-MUSCAT simulations and SEVIRI pixel data. How you combine these data and how you overcame the different spatial resolution of the compared data sets?

2. Do you believe that your results (Figure 7) can be applicable in other regions inside the Meteosat domain (e.g the European side of the Mediterranean, where dust transportation from Sahara, occurs?). Please describe your analytic opinion in the document.

Specific comments • Page 4 (Equations 1-3): Please, refer that the variables "Min" and "Max" represent Brightness Temperature (BT) values. Also, describe how these variables obtain their values because it is not clear (last paragraph of the section 2.1) mainly when you refer that there is a case that BT can be larger that the maximum values (page 4, lines 14-20, just below the relative equations).

• Table 3 and Table 4: Please, refer what represent the values of the columns with the SEVIRI spectral channel centers (98.7, 10.8, 12.0 $\mu$m) and which is the unit (it is not clear). Please, check anywhere else may also needed.

---

## Author Comment (AC1) · 15 Mar 2019

Responses to reviews of "The sensitivity of the colour of dust in MSG-SEVIRI Desert Dust infrared composite imagery to surface and atmospheric conditions"

We thank both of the reviewers for their helpful comments, and for their interest in our work. Below are our responses to their comments, with their comments included in bold and our additions to the text of the manuscript included in italics.

**General comments 1. There is no relative information about the spatiotemporal correlation between COSMO-MUSCAT simulations and SEVIRI pixel data. How you combine these data and how you overcame the different spatial resolution of the compared data sets?**

1) With the exception of Figure 1 (two indroductory SEVIRI RGB images) and the SEVIRI channel spectral response functions, used for calculating the absorption and scattering properties of the simulated dust, no SEVIRI data are used in this study. All of the colours in Figures 6 and 7, along with the transmittances in Tables 3 and 4, are derived from the COSMO-MUSCAT-RTTOV simulations. Hence all of the analysis in both Sections 3 and 4 is performed at the model resolution of 0.25° and no spatiotemporal matching between the simulations and the SEVIRI pixels is required.

**General comments 2. Do you believe that your results (Figure 7) can be applicable in other regions inside the Meteosat domain (e.g. the European side of the Mediterranean, where dust transportation from Sahara, occurs?). Please describe your analytic opinion in the document.**

2) Although the Desert Dust RGB imagery is only designed for observing dust over the desert, Figure 7 does provide hints as to what we would expect to see over Europe and the oceans, both of which can be classified within the high emissivity zone as defined in this paper. Europe is likely to be classified to be within the high emissivity zone, and with cooler skin temperatures. Looking at the relevant segments in Figure 7 it is apparent that such dust may be perceptible over Europe, but will be harder to see against the background surface.

To emphasise the difficulty of observing dust over Europe using this style of imagery, we have also performed simulations over Europe. For reference, the comparable plot for Europe ( $>36^{\circ}N$ ) is shown at the end of this response document. The maximum
AOD within this latitude range during this time period is 1.31, so the four outer dust rings in this plot consist only of points with an AOD of greater than 1. The maximum skin temperature is 327.3 K, hence the 'Very hot' skin temperature range has been removed and replaced by the 'Cold' range. The minimum emissivity at 8.7  $\mu$ m is 0.945.

In the conclusions, we have added the following paragraph:

"The focus of this paper has been on the perceptibility of dust in the Desert Dust imagery over North Africa in summer, excluding Europe, the oceans, and other seasons of the year. It should be stressed that the Desert Dust imaging scheme is not designed to resolve dust over non-desert regions. However, the insights produced by Figure 7 are also applicable to remote regions, since physically the same variables are relevant for the measured/simulated signal at TOA. Europe and the oceans typically have surfaces characterised by higher emissivities and cooler temperatures compared to North Africa. Identifying the relevant segments in Figure 7 (high-cool), it is to be expected that sufficiently thick dust can be observed over Europe and the oceans, however due to the enhanced background values of the green beam such dust will be less apparent compared to over North Africa. Meanwhile the other seasons of the year will have cooler skin temperatures, in which regimes there are weaker colour contrasts between the dust and the background surfaces."

Page 4 (Equations 1-3): Please, refer that the variables "Min" and "Max" represent Brightness Temperature (BT) values. Also, describe how these variables obtain their values because it is not clear (last paragraph of the section 2.1) mainly when you refer that there is a case that BT can be larger that the maximum values (page 4, lines 14-20, just below the relative equations).

3) In the last paragraph of Section 2.1 we have now added the sentence: "*Minima and maxima refer to the brightness temperature or BTD values such that, for example, a*  $T_{B120} - T_{B108}$  difference of -4 K gives rise to a red value of 0, and a difference of +2 K gives rise to a red value of 1." We hope that this clarifies the formation of the RGB
caption to Table 1.
Table 3 and Table 4: Please, refer what represent the values of the columns with the SEVIRI spectral channel centers (98.7, 10.8, 12.0  $\mu$ m) and which is the unit (it

4) We have added the words "*Transmittances (unitless)*" above the 8.7, 10.8 and  $12.0 \,\mu$ m columns in both Tables 3 and 4. We hope that this addition clarifies the Tables. The fact that extinction efficiences are unitless has also been included in the

is not clear). Please, check anywhere else may also needed.

**Anonymous Referee 2**

General comments. My only concern, apart from some minor specific comments listed below, is that the scope of the study should be better defined and motivated. The study is confined to northern African land, but Saharan dust is transported over the Atlantic Ocean and Europe. What about the capabilities to detect dust in these regions, where the surface and atmospheric conditions (as well as the satellite viewing angles) are markedly different? The study is furthermore confined to two summer months. What about the rest of the year?

1) The Desert Dust scheme is only designed to observe dust over deserts, exploiting the emissivity contrast between dust and the desert surface. Other surfaces as are typically found in Europe, central Africa, and in mountainous regions, do not have this contrast. This contrast is similarly weak over the ocean. Hence it is not recommended to make substantial use of the Desert Dust imagery over these regions remote from the dust sources, instead other imaging schemes which make use of visible channels tend to be more appropriate. We include the following text in the third paragraph, splitting off the "*B2018...*" into a separate paragraph:

"The scheme was specifically designed to discriminate dust over desert surfaces: during the daytime there is a particularly strong contrast in colour in the composite imagery between the deep pink colours exhibited by dust storms and the light blue colours characteristic of hot desert surfaces. This technique exploits the contrast in the thermal emissivity properties of finer lofted dust particles with respect to the coarser particles composing the desert surface (e.g. Wald et al., 1998). Over surfaces remote from the dust source regions, such as the vegetated regions characteristic of Europe and central Africa, and over the oceans, the emissivity contrast between lofted dust and the Earth surface is much reduced and it is therefore much harder to distinguish dust over these regions. Over such regions detection and quantification methods using solar channels are likely to be more appropriate (e.g. Kaufman et al., 1997; Brindley & Ignatov, 2006). This paper is focused on the dust source regions of
**North Africa for which the Desert Dust scheme has been optimised."**

We have also included an extra paragraph in the Conclusions discussing the possibilities for observing dust over oceans and over Europe, see point 2) in our reply to Anonymous Referee 1. The fact that the colours may be different in other seasons of the year is also discussed in this paragraph, pointing out that the other seasons would be expected to have cooler skin temperatures: the cooler regimes in Figure 7 would provide information about the colours that we would expect to observe in the other seasons. This was also discussed in the following sentence at the end of the second paragraph of Section 4.2:

"This description generalises the 'night-time effect', of redder background colours during the night, as being rather a 'cool skin temperature effect' which is applicable not just to the diurnal but also to the seasonal cycle."

**Page 2, line 4-8: Can you mention a concrete example of a feedback loop?**

2) We now include in the text the following sentence at the end of the first paragraph: "For example, the direct radiative effect of dust modifies the atmospheric stratification and synoptic scale pressure patterns, which in turn has an impact on dust mobilisation and transport (Heinold et al. 2011a)."

**Page 3, line 6-8: It is not clear to me what the significance of this statement is for the current paper.**

3) This is a fair point, the sentence just describes a finding from the previous paper but it has no particular importance for the current paper, so we are happy to remove this sentence.

**Page 5, line 9: Is the solar zenith angle relevant here?**

4) It is not relevant for the infrared simulations in RTTOV, so we have removed this from the sentence.
Page 5, line 14-15: I understand that you focus on cloud-free scenes, but it would be useful to say a few words about the signature of clouds in the imagery. And what about scenes with both clouds and dust?

5) Cloud has very strong signatures in the Desert Dust imagery, so towards the end of Section 2.2 we now include the following sentence:

"Clouds produce divergent signatures in the Desert Dust imagery depending on their height, thickness, and composition, giving rise to dark red colours in the case of thick cumulonimbus clouds (Lensky and Rosenfeld, 2008)."

Scenes with cloud and dust add an extra layer of complication. Unless the cloud layer is very thin, then dust under cloud will almost always be obscured by the overlying cloud. This is a particular problem for identifying dust storms originating from haboobs, convective systems which can be responsible for some of the thickest dust events. Sufficiently thick dust may be observable over a low altitude cloud layer, but the resulting changes in the apparent background surfaces due to the cloud makes identification of the dust itself a challenging task.

**Page 7, line 34: What does 'these' refer to?**

6) 'These' refers to the channel emissivity values, now clarified in the text.

Page 10, line 20-22: Can you explain why the thickest dust storms are located low in the atmosphere while lower loadings occur more frequently at higher elevations? Is it simply because the thickest loadings occur close to the (storm) source region and so have not had time to reach higher altitudes?

7) This is correct, this is simply because the thickest loadings occur closest to the source regions. We include the following sentence in the last paragraph of Section 3.5: "*Dust loadings are largest close to the source regions, and on transport to remote regions the dust concentrations are diluted both vertically and horizontally.*"

**Page 16, line 21: The imager on MTG (FCI) will have somewhat different infrared channels than SEVIRI. For the three channels used in the desert dust RGB, the central wavelengths will move from 8.7, 10.8, and 12.0 micron to 8.7, 10.5 and 12.3 micron, respectively. Is there something to say about whether and how this will impact the detectability of dust?**

8) Indeed this change in the channel wavelengths for the MTG FCI will have an impact on the Desert Dust imagery, assuming the same configuration of channels is used for the FCI dust imagery. Shifting the channels at 10.8 and 12.0  $\mu$ m apart to 10.5 and 12.3  $\mu$ m would increase the spectral contrast between the dust extinction of the two channels, leading to redder and therefore more vivid pink dust colours. Meanwhile the shift upwards from 12.0 to 12.3  $\mu$ m would increase the atmospheric absorption under low aerosol loadings, suppressing the red beam with respect to SEVIRI and therefore the background desert colours would appear bluer in the FCI imagery. However, we do not believe that a discussion on this fits into the relevant paragraph in the Conclusions.

**Page 26: Could you add in the caption of Fig. 2 what the white patches correspond to. I assume there is no data because of nearby water surfaces, but there are also some white patches in the middle of the Sahara. Are these oases?**

9) In Figures 2-4 the white patches correspond to grid cells which contain some fraction of water, which may indeed be surprising for various points in the central Sahara. As an example, the white patch in southern Libya is in the vicinity of the Waw an Namus area, an area which is of volcanic origin and contains a number of small lakes. Due to these lakes the land area fraction of the corresponding grid cells is very slightly less than 1. In our analysis we maintain a strict criterion that the land area fraction for a grid cell must be equal to 1 for an RTTOV simulation to be performed on it, given the differences in the thermal emissivity and surface temperature properties between land and water. Hence in the caption of Figure 2 we now include the sentence: "White
regions correspond to grid cells which contain some fraction of water." Related to this, Figures 2(a-d), 3 and 4 have been updated from the previous contour plots, the colours presented are now the discrete colours for each grid cell. The differences are very minor indeed, but this helps to make the plots more precise.

---

## Author Response (AR2)

**Responses to Co-editor decision on "The sensitivity of the colour of dust in MSG-SEVIRI Desert Dust infrared composite imagery to surface and atmospheric conditions"**

Thank you for your response, and for your comments, which we have now acted on. Below are our responses, and overleaf is the marked-up version of the updated manuscript.

**1. In the Abstract, please remove the reference made to the paper by Banks et al. (2018). Please, rephrase the relevant sentence.**

This reference has been removed, and so the relevant sentence has been amended appropriately. This has also necessitated a re-write of the subsequent sentence.

**2. In lines 26-27 of page 16, the meaning of the relevant sentence is not clear, please re-write it.**

This sentence has now been split into two sentences and re-written slightly, we hope that this makes the meaning clearer.

**3. In the last paragraph of Conclusions, page 17, please avoid writing in first person (plural), rather write in third person.**

This paragraph has been changed so that there are no incidences in the text of the first person.

Further to these, we have also included a more recent reference to RTTOV on p.5, in addition to those previously cited.

[revised manuscript text omitted]